# Entomopathogenic Fungi-Mediated Solubilization and Induction of Fe Related Genes in Melon and Cucumber Plants

**DOI:** 10.3390/jof9020258

**Published:** 2023-02-15

**Authors:** Fabián García-Espinoza, Enrique Quesada-Moraga, María José García del Rosal, Meelad Yousef-Yousef

**Affiliations:** 1Departamento de Agronomía (DAUCO María de Maeztu Unit of Excellence 2021–2023), Campus de Rabanales, Universidad de Córdoba, Edif. C4, 14071 Cordoba, Spain; 2Departamento de Parasitología, Universidad Autónoma Agraria Antonio Narro–Unidad Laguna, Periférico Raúl López Sánchez S/N, Torreón 27054, Coahuila, Mexico

**Keywords:** Entomopathogenic fungi, growth promoters, nutrient solubilization, bioavailability, iron acquisition genes, ferric reductase activity

## Abstract

Endophytic insect pathogenic fungi have a multifunctional lifestyle; in addition to its well-known function as biocontrol agents, it may also help plants respond to other biotic and abiotic stresses, such as iron (Fe) deficiency. This study explores *M. brunneum* EAMa 01/58-Su strain attributes for Fe acquisition. Firstly, direct attributes include siderophore exudation (in vitro assay) and Fe content in shoots and in the substrate (in vivo assay) were evaluated for three strains of *Beauveria bassiana* and *Metarhizium bruneum*. The *M. brunneum* EAMa 01/58-Su strain showed a great ability to exudate iron siderophores (58.4% surface siderophores exudation) and provided higher Fe content in both dry matter and substrate compared to the control and was therefore selected for further research to unravel the possible induction of Fe deficiency responses, Ferric Reductase Activity (FRA), and relative expression of Fe acquisition genes by qRT-PCR in melon and cucumber plants.. In addition, root priming by *M. brunneum* EAMa 01/58-Su strain elicited Fe deficiency responses at transcriptional level. Our results show an early up-regulation (24, 48 or 72 h post inoculation) of the Fe acquisition genes *FRO1*, *FRO2*, *IRT1*, *HA1*, and *FIT* as well as the FRA. These results highlight the mechanisms involved in the Fe acquisition as mediated by IPF *M. brunneum* EAMa 01/58-Su strain.

## 1. Introduction

Insect pathogenic fungi (IPF), which are among the most important biological control agents to be commercially developed for the management of a wide range of chewing and piercing/sucking insect pests, have multifunctional lifestyles and can interact with crops as endophytes establishing mutualistic interactions that benefits the host plant e.g., enhanced plant growth, development, immunity and resistance to biotic and abiotic stresses [1,2,3]. IPF can dwell internally in plant tissues including competence in the rhizosphere eliciting no disease symptoms in the plant while targeting insect pests even providing systemic protection of the plant against insect pests and contributing to increased plant growth [2,4,5]. The genera *Beauveria* and *Metarhizium* are among the most studied IPF [1,6] and are considered excellent examples of fungi with multifunctional lifestyles [7].

In recent works, IPF have been shown to be involved in plant acquisition of nutrients [8,9] and plants grown in the presence of fungal partners exhibit increased growth and productivity [7], e.g., Plant inoculation with *M. brunneum*, *B. bassiana* and *Isaria farinosa*, has significant effects on growth and development of some important crops such as sorghum, wheat, sunflower and tomato [9,10,11]. Besides this, *M. brunneum* increased Fe availability on calcareous soil and alleviate Fe chlorosis in sorghum wheat and sunflower plants [10,12] as well as crop protection against microbial pathogens [7].

A lack of iron (Fe) is considered one of the major crop productivity constraints worldwide [13]. Fe is a micronutrient that is essential for a range of important enzymatic processes in most organisms and in most environments Fe deficiency is not triggered by low total Fe concentrations but by low Fe bioavailability [14]; to over-come these limitations, bacteria, fungi, and gramineous plants (grasses) are known to sequester Fe using siderophores [15]. A siderophore is a low-molecular-weight Fe (III) ligand and they function as biogenic chelators with high affinity and specificity for Fe complexes [16].

According to Winkelmann [17], both fungi and plants, unlike bacteria, are immobile organisms, therefore, to grow, both groups depend on local conditions and concentrations of nutrients, this also applies to ferric nutrition that can be improved by the secretion of siderophores and organic acids for the demineralization of other nutrients; foraging generally occurs at the tips of the growing hyphae, that is, through the propagation of the mycelium they are able to explore and exploit the resources of their environment.

Under Fe deficiency conditions, plants develop morphological and physiological responses, mainly in their roots, aimed to facilitate its acquisition [13,18,19,20]. The main physiological responses are: enhanced ferric reductase activity; enhanced Fe^2+^ transport; rhizosphere acidification; and increased synthesis and/or release of organic acids, phenolic compounds, such as coumarins, and flavins, which can act as chelating and reducing Fe agents, improving its solubility for plants [21,22,23,24,25,26,27]. The main morphological responses are aimed to increase the contact surface of roots with soil and include development of subapical root hairs; of cluster roots (also named proteoid roots); and of transfer cells [28,29,30,31].

In the regulation of the Fe deficiency responses hormones and regulating substances such us ethylene and nitric oxide (NO) have been involved, which act as positive regulators [32,33]. Ethylene and NO exert their function through *FIT*, a bHLH transcription factor (TF) which interacts with other TFs such as bHLH38, bHLH39, bHLH100 and bHLH101 [34,35,36]. All of them increase their expression under Fe deficiency conditions [37]. Besides bHLHs, *FIT* also interacts with MYB72 and MYB10, two other TFs essential for plant growth on low Fe conditions [38,39,40].

The IPF *Beauveria caledonica* has shown efficacy not only in solubilizing and transforming toxic minerals, but also in tolerating and thriving on them [41] and the IPF *Metarhizium robertsii* has been shown to produce a complex of extracellular siderophores, including N^α^-dimethylcoprogen (NADC) and dimerumic acid (DA) when it is cultivated under iron-depleted conditions [15]. Some reports indicate that *B. bassiana* is a good producer of siderophores [11] while others suggest that some species of *Metarhizium* are not [42]. Compounds secreted by microorganisms may in turn help to improve the solubility of Fe in soils and plant Fe nutrition via elevated microbial activity [43]. A remarkable fact is that fungi, unlike bacteria, can avoid competition for nutrients with plants [44], however, there are no studies on the mechanisms used by IPF for Fe acquisition by plants. Hereby, direct and indirect mechanisms of IPF alleviation of Fe chlorosis in cucumber and melon plants have been investigated.

## 2. Materials and Methods

### 2.1. Fungal Isolates and Inoculum Preparation

Two isolates of *B. bassiana* (EABb 04/01-Tip and EABb 01/33-Su) and one isolate of *M. brunneum* (EAMa 01/58-Su) from the culture collection of the Agronomy Department, University of Cordoba (Spain) were used in the experiments (Table 1). Transient and temporary endophytic colonization of melon plants has been previously demonstrated by foliar application of these isolates [45,46].

To provide inoculum for experiments, all isolates were subcultured from stored slant cultures on Potato Dextrose Agar (PDA) in Petri dishes and grown for 15 d at 25 °C in darkness.

### 2.2. In Vitro Study of Fe Biodisponibility by Production of Siderophores

The in vitro study was done to investigate the abilities of fungal isolates to demineralize Fe. Prior to the test, isolates were grown in Potato Dextrose Agar (PDA) medium to obtain four-day old mycelium. This assay was repeated twice with four biological replicates per isolate.

We followed a simplified method [47] of the universal chemical assay for siderophores detection [16], with FeCl_3_ is used as FeIII source. Discs (6 mm diameter) of mycelium from each isolate (6 mm/myc) were cut from actively growing colonies (4 d) and placed at the center of Petri plates (9 cm) containing Chrome Azurol Sulfonate (CAS) agar medium. Plates were incubated at 26 (±2) °C in darkness for 10 d [11]. Daily from 3–10 days post inoculation (dpi) both the diameters of colonies and areas of yellow/orange halo surrounding them were measured from photographs taken using the software ImageJ (National Institute of Health, Bethesda, MD, USA); the size of the orange-coloured area was indicative of the quantity of siderophores produced [48].

### 2.3. In Planta and Soil Studies of Fe Biodisponibility

To evaluate Fe acquisition in melon plants, a completely randomized design with 3 treatments (3 strains applied by soil drenching), and their respective control, with 6 replicates (plants) per treatment were used.

The substrate (Floragard, Germany) was sterilized twice in an autoclave (121 °C for 30 min), with an interval of 24 h [49]. The pots with a capacity of 500 mL, previously washed and sterilized, were filled with the sterilized substrate. Certified endophyte-free melon (*Cucumis melo* L. cv. Galia) was used as crop in all experiments, as in our previous studies [45,50]. Seeds were surface sterilized according to Garrido-Jurado et al. [46].

Inoculum preparation was carried out by scraping the conidia from the Petri plates into a sterile solution of 0.1% Tween 80, followed by sonication for 5 min to homogenize the inoculum and filtration through several layers of cheesecloth to remove any mycelia.

A hemocytometer (Malassez chamber; Blau Brand, Wertheim, Germany) was used to estimate conidia concentration which was finally adjusted to 1 × 10^8^ conidia/mL by adding a sterile solution of distilled water with 0.1% Tween 80.

Soil drenching was carried out when the melon plants reached four true leaves stage, 30 d after seedling; 5 mL of the suspension was poured with a pipette onto the surface of the pot. Control plants were treated similarly with a sterile solution of 0.1% Tween 80. Then, at 50 dpi, elemental analysis in dry matter and substrate was carried out. For that, the substrate and vegetal material, including aerial parts and roots were dried in an oven at 60 °C for 96 h and weighed.

The content of Fe in dry matter and substrate was evaluated using the modified “Olsen Phosphorus” technique [51]. For that, both dry matter and substrate was grinded to obtain a homogeneous mixture, then, 0.2 g of sample per replicate per treatment was added to a 100 mL precipitate glass; in a vapor extraction hood, 3 mL of nitric acid (65%) were added and covered with a watch glass, 16 h after, 1 mL of perchloric acid (70%) was added to each glass [52,53]. Fe was determined with an atomic absorption spectrophotometer (Perkin–Elmer Analyst 200).

### 2.4. Ferric Reductase Activity and Fe Acquisition Gene Expression

#### 2.4.1. Growth Conditions and Vegetal Material

To study the activity of the ferric reductase and the relative expression of the Fe acquisition genes we used two species of cucurbits (*Cucumis melo* L. var. Futuro and *Cucumis sativus* L. var Ashley, Semillas Fitó, S.A., Barcelona, Spain).

Plants were grown under controlled conditions as previously described [54]. Briefly, seeds of both species were sterilized with 5% HCl for 5 min, stirring constantly, then washed twice with sterilized water and placed on absorbent paper moistened with 5 mM CaCl_2_, covered with the same paper and placed at 25 °C in the dark over 3 days for germination. Then, when the plants sufficiently elongated their stems, they were transferred to a hydroponic system culture that consisted of a thin polyurethane raft with holes on which plants inserted in plastic lids were held floating on the aerated nutrient solution. Plants grew in a growth chamber at 22 °C day/20 °C night temperatures, with relative humidity between 50 and 70%, and a 14-h photoperiod at a photosynthetic irradiance of 300 μmol m^−2^ s^−1^ provided by white fluorescent light (10.000 lux).

The nutrient solution used was R&M [55] whose composition is the following: macronutrients: 2 mM Ca(NO_3_)_2_, 0.75 mM K_2_SO_4_, 0.65 mM MgSO_4_, 0.5 mM KH_2_PO_4_, and micronutrients: 50 μM KCl, 10 μM H_3_BO_3_, 1 μM MnSO_4_, 0.5 μM CuSO_4_, 0.5 μM ZnSO_4_, 0.05 μM (NH_4_)6Mo_7_O_24_, and 10 μM Fe-EDDHA.

After 10 days (in the case of cucumber) and 13 days (in the case of melon) of growth, plants were separated into four groups that posteriorly constituted the 4 treatments, as described below.

#### 2.4.2. Inoculum Preparation and Roots Priming

*Metarhizium brunneum* (EAMa 01/58-Su strain) was chosen to be used in this part of the study due to the properties previously shown to solubilize Fe. Inoculum was prepared as previously described and adjusted to 1 × 10^7^ conidia/mL by adding sterile solution of distilled water with 0.1% Tween 80.

Plants with two true were selected and placed in trays with 2.5 L of inoculum solution. Control plants (un-inoculated) were placed in trays with 2.5 L of 0.1% Tween 80. All plants were maintained in continuous agitation for 30 min. After that, inoculated and un-inoculated plants were transferred to two different nutritional conditions, Fe sufficient (+ Fe40µM) and deficient (– Fe) so that finally four treatments with 42 plants were used: Control + Fe40µM (un-inoculated), Inoculated + Fe40µM, Control—Fe (un-inoculated), Inoculated—Fe. Each assay with both species of *Cucumis* was repeated twice.

#### 2.4.3. Measure of Ferric Reductase Activity (FRA)

The FRA was determined as described by García et al. [56]. Previously to determine FRA, plants were subjected to a pre-treatment for 30 min in plastic vessels with 50 mL of a nutrient solution without micronutrients, pH 5.5. Then they were transferred into 50 mL of a Fe (III) reduction assay solution for 1 h. This assay solution consisted of nutrient solution without micronutrients, 100μM Fe(III)-EDTA and 300 μM Ferrozine, pH was adjusted to 5.0 with KOH. The environmental conditions during the measurement of Fe (III) reduction were the same as the growth conditions described above. FRA was determined spectrophotometrically by measuring the absorbance (562 nm) of the Fe(II)-Ferrozine complex and by using an extinction coefficient of 29.800 M^−1^ cm^−1^. After that, roots were excised and weighed, and the results were expressed on a root fresh weight basis. Also, SPAD values (as a proxy of the chlorophyll concentration in leaf) were measured daily with a portable chlorophyllmeter (SPAD 502 Minolta Camera Co., Osaka, Japan).

#### 2.4.4. RNA Isolation, cDNA Synthesis and qRT-PCR Analysis

Real-time PCR analysis was carried out as described by García et al. [19]. Briefly, roots and true leaves were ground to a fine powder with a mortar and pestle in liquid nitrogen. Total RNA was extracted using the Tri Reagent solution (Molecular Research Center, Inc., Cincinnati, OH, USA) according to the manufacturer’s instructions. cDNA synthesis was performed by using iScript^TM^ cDNA Synthesis Kit (Bio-Rad laboratories, Inc., Hercules, CA, USA) from 3 µg of DNase-treated RNA as the template and random hexamers as the primers. As internal control 18S cDNA was amplified using the QuantumRNA Universal 18S Standards primer set (Ambion, Austin, TX, USA); the thermalcycler program was one initial cycle of 94 °C for 5 min; followed by cycles of 94 °C for 45 s; 55 °C for 45 s; 72 °C for 1 min, with 27–30 cycles, all followed by a final 72 °C elongation cycle of 7 min [32,33,54,57].

The study of gene expression by qRT-PCR was performed in a qRT-PCR Bio-Rad CFX connect thermal cycler and the following amplification profile: initial denaturation and polymerase activation (95 °C for 3 min), amplification and quantification repeated 40 times (94 °C for 10 s, 57 °C for 15 s and 72 °C for 30 s), and a final melting curve stage of 65 °C to 95 °C with increment of 0.5 °C for 5 s to ensure the absence of primer dimer or non-specific amplification products. PCR reactions were set up with 2 µL of cDNA in 23 µL of SYBR Green Bio-Rad PCR Master Mix, following the manufacturer’s instructions [19,58]. Standard dilution curves were performed for each primer pair to confirm appropriate efficiency of amplification (E = 100 ± 10%). Relative expression of *FRO1, IRT1* and *HA1* were studied in *C. sativus* while *FRO1*, *FRO2*, *FRO3*, *FRO4*, *IRT1* and *FIT* were studied in *C. melo*. Constitutively expressed *ACTIN* [59] and *CYCLO* genes, were used as reference genes to normalize qRT-PCR results. Table 2 contents the list of primers that were used in this study. The relative expression levels were calculated from the threshold cycles (Ct) values and the primer efficiencies by the Pfaffl method [60]. Each PCR analysis was conducted on three biological replicates and each PCR reaction repeated twice.

### 2.5. Data Analysis

Iron siderophore production data, total and relative content of Fe in dry matter and substrate and data of FRA were analyzed using analysis of variance (ANOVA) followed by a Tukey multiple range test, different letters over the bars indicate significant differences (*p* < 0.05) among treatments (Statistix 9.0^®^, Analytical Software, Tallahassee, FL, USA).

Results of relative gene expressions were analyzed using one-way analysis of variance (ANOVA) followed by a Dunnett’s test, * (*p* < 0.05), ** (*p* < 0.01) or *** (*p* < 0.001) over the bars indicate significant differences in relation to the control treatment (GraphPad Prism 9.4.0, GraphPad Software, LLC, 2365 Northside Dr., Suite 560, San Diego, CA 92108 USA). Data of gene expression represent the mean of three independent technical replicates.

## 3. Results

### 3.1. Iron Siderophores Exudation

There were significant differences amongst isolates in siderophore production 10 dpi (*F*_2,21_ = 117.73, *p =* 0.000); *M. brunneum* isolate EAMa 01/58-Su was the most capable of changing the largest area of CAS agar from blue to orange (58.4%), while *B. bassiana* isolates EABb 04/01-Tip and EABb changed the color of only 24.35% and 17.88%, respectively (Figure 1). The timeline for Fe siderophores exudation shown in Figure 3 reveals the difference between the *M. brunneum* isolate and the others from 3 dpi onwards.

### 3.2. Total Dry Matter and Fe Content in Dry Matter and Substrate

Significant differences were observed on dry matter when we compared EAMa 01/58-Su (*F*_1,8_ = 10.63, *p* = 0.0115), EABb 04/01-Tip (*F*_1,8_ = 5.88, *p* = 0.0416) and EABb 01/33-Su (*F*_1,8_ = 6.78, *p* = 0.0314) treatments vs. control, however we can see that plants inoculated with EAMa 01/58-Su produced the highest dry matter content (Figure 2A). On another hand, no significant differences were observed on Fe content in dry matter when we compared each treatment vs. control [(*F*_1,8_ = 2.68, *p* = 0.1400), (*F*_1,8_ = 2.08, *p* = 0.1870), (*F*_1,8_ = 3.0, *p* = 0.1213), for EAMa 01/58-Su, EABb 04/01-Tip and EABb 01/33-Su, respectively] (Figure 2B). In the case of relative Fe content in the substrate, only EAMa 01/58-Su treatment vs. control presented significant difference (*F*_1,6_ = 7.77, *p* = 0.0317) (Figure 2C); there weren’t significant differences between EABb 04/01-Tip (*F*_1,6_ = 3.41, *p* = 0.1143) and EABb 01/33-Su (*F*_1,6_ = 0.37, *p* = 0.5629) treatments when were compared vs. control (Figure 2C).

### 3.3. Ferric Reductase Activity and Genes Responsible of the Reduction and Transport of Iron

In general, FRA presented higher values in cucumber and melon plants grown under Fe deficient conditions. In cucumber plants, reductase activity was higher in Fe deficient plants inoculated with *M. brunneum* (EAMa 01/58-Su strain) in comparison with their respective controls over the seven days of the study (Figure 3A). However, significant differences were detected between Fe deficient cucumber inoculated and un-inoculated at 4, 5 and 7 dpi [(*F*_3,22_ = 13.68, *p* = 0.0001), (*F*_3,20_ = 35.3, *p* = 0.0000) and (*F*_3,19_ = 74.68, *p* = 0.0000), respectively] (Figure 3A). In the case of melon, significant differences were found between Fe deficient plants inoculated at 3 dpi relative to the un-inoculated Fe deficient plants (*F*_3,20_ = 61.23, *p* = 0.0000) (Figure 4A).

**Figure 3 jof-09-00258-f003:**
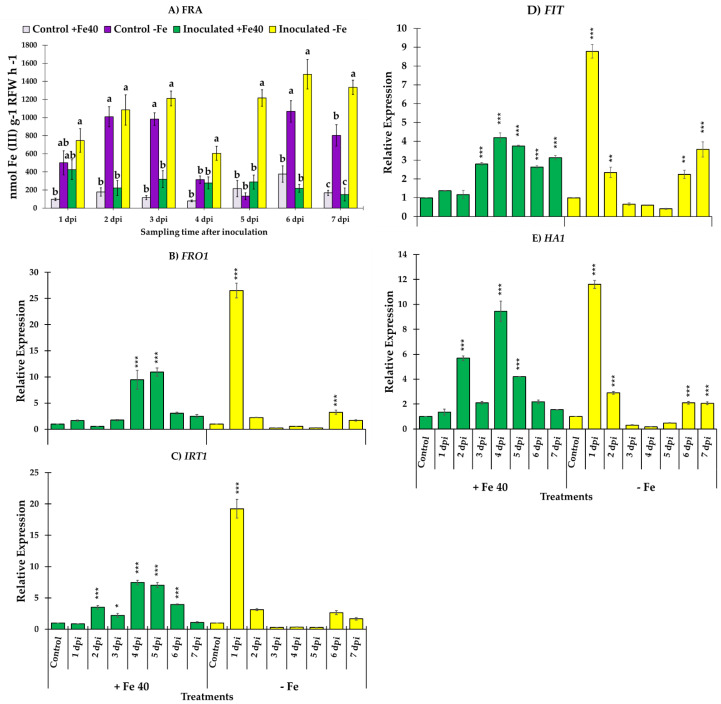
Evolution of FRA along seven days of observation and relative expression of *FRO1*, *IRT1*, *FIT* and *HA1* in *C. sativus* roots. Four treatments were used, namely, (i) Control + Fe40µM (un-inoculated), (ii) Inoculated + Fe40µM, (iii) Control - Fe (un-inoculated) and (iv) Inoculated - Fe. The expression of control treatment for each nutritional condition is presented once, at the beginning of the graph, with the relative expression comparison method used, the control is always equal to 1. Data of gene expression represent the mean of three independent technical replicates, according to the Dunnett’s test, * (*p* < 0.05), ** (*p* < 0.01) or *** (*p* < 0.001) over the bars indicate significant differences in relation to the control treatment. In the case of FRA, letter over the bars denote significant difference between plants inoculated and control plants analyzed by completely randomized ANOVA followed by a Tukey test (*p* < 0.05).

Relative expression levels of Fe acquisition genes, *FRO1*, *IRT1*, *FIT* and *HA1* in cucumber are represented in Figure 3B–E. Fe acquisition genes experimented an increase of their expression levels after the inoculation with *M. brunneum* EAMa 01/58-Su strain in both conditions, Fe sufficient and deficient, in comparison with their respective un-inoculated controls at different times (Figure 3B–E). However, the relative expression levels of *FRO1*, *IRT1*, *FIT* and *HA1* reached at the first day post inoculation were much higher in Fe deficient conditions than Fe sufficient, being this increment of 26, 19, 8.8 and 11 times to *FRO1, IRT1, FIT* and *HA1* respectively (Figure 3B–E). In Fe sufficient conditions we observed an increase of the relative expression genes studied at different times post inoculation but in any cases the values reached were like that observed in Fe deficient conditions.

Generally, the results obtained in melon were like the ones obtained in cucumber. In this case we had the possibility to study three different genes that codify ferric reductase enzymes *FRO1, FRO2* and *FRO3* besides *IRT1* and *FIT*. As occur in cucumber roots, the relative expression of all genes studied was higher in Fe deficient conditions except in the case of *IRT1,* in which no significant differences were found in the relative expression values between Fe sufficient and deficient conditions (Figure 4E). *FRO1*, *FRO3* and *FIT* reached its maximum relative expression value at the second day post inoculation (Figure 4B,D–F) while *FRO2* did it on the third day and *IRT1* on the sixth (Figure 4C,E). Although *IRT1* reached its maximum relative expression level later, it also experimented a significant increase at the second day after inoculation as the rest of genes (Figure 4E).

Relative expression of two ferric reductase genes, *FRO3* and *FRO4,* involved in Fe^3+^ reduction in leaves were also studied in melon plants. *FRO3* and *FRO4* relative expression significantly increased at the first day post inoculation in Fe deficient conditions. However, in the case of *FRO4* the maximum relative expression level reached occur at the second day post inoculation in Fe deficient conditions. As occur with the genes studied in roots, in Fe sufficient conditions no significant differences were observed after inoculation except at the first day post inoculation in *FRO4* where a significant increase was observed (Figure 5).

In Figure 6 and Figure 7 it is represented a panorama of FRA and general appearance of aerial parts and roots at 5 dpi. Both, cucumber (Figure 6) and melon (Figure 7) plants, began to show deficiency symptoms at 4 dpi, being more visible in the cucumber plants, where leaves with a higher degree of chlorosis were observed. In both species, the roots of the plants that grew with sufficient Fe had a more elongated appearance and less abundant secondary roots as it can be seen in the picture. Also, in cucumber plants, SPAD values from 4 to 7 dpi, have shown to be significantly different between treatments (*F*_3,95_ = 42.11, *p* = 0.0000), especially in those grown under Fe deficient conditions (Figure 6B), nonetheless, plants grown in Fe sufficient conditions show higher chlorophyll content; in the case of melon, inoculated plants grown under Fe deficient conditions, were those that presented higher chlorophyll content with significant difference respect to other treatments (*F*_3,85_ = 14.89, *p* = 0.0000) (Figure 7B).

## 4. Discussion

The discovery of new functions for IPF as plant endophytes and growth promoters, and their competence in the rhizosphere have enabled the expansion of their use, thus providing added value to their main use as biological control agents against a wide variety of insects and mites harmful to cultivated plants [2,50]. In this sense, many studies have shown that IPF represent an excellent alternative to control agricultural pests [8,11,63,64,65,66]. Indeed, several studies have shown the efficacy of species from the genera *Metarhizium* and *Beauveria* to control herbivores in crops like olive, corn, wheat, tomato, sunflower, melon and soybean amongst others [11,49,65,67]. Besides, they play other roles beyond pest control with direct and indirect benefits for plant growth through nutrient mobilization and/or mediation of trophic relationships [8,9,68,69,70]. Increasing the bioavailability of nutrients through phytohormones production and improvement of water transport are ways that IPF promote plant growth directly; they also benefit plants through indirect mechanisms involving induction of systemic resistance to harmful organisms [11].

However, little is known about direct and indirect mechanisms used by IPF for Fe acquisition in plants, although many studies indicated that IPF alleviate Fe chlorosis symptoms as in previous studies [9,12], *M. brunneum* EAMa 01/58-Su was also the best growing in culture medium with low Fe availability. In the same way, Raya-Díaz et al. [9] showed that *M. brunneum* EAMa 01/58-Su applied to the soil at high doses (5 × 10^8^ conidia ml^−1^) alleviated Fe chlorosis symptoms in sorghum plants grown in calcareous soil, and increased plant height and inflorescence production of sunflowers grown in calcareous and non-calcareous soils.

Our in vitro study demonstrated the ability of *M. brunneum* isolate EAMa 01/58-Su to demineralize Fe being the most effective in producing Fe siderophores, with 58.4% of surface siderophores exudation 10 dpi, while *B. bassiana* isolates EABb 04/01-Tip and EABb 01/33-Su only achieved 24.3% and 17.8% of surface siderophores exudation, respectively. The increase of Fe availability resulting from application of a specific isolate could either be due to secretion of organic acids, thus reducing the pH of the medium, or through release of siderophores that chelate not only Fe but also other nutrients such as Zn, Mn and Cu [9,15]. There are few reports about IPF activity as solubilizers of nutrients. Some studies showed similar data using the well-known genus *Trichoderma* [71] and others using the saprophyte *Aspergillus niger* showing abilities as phosphorus solubilizers [72,73,74,75]. Recent studies by Barra-Bucarei et al. [11] showed differences between five isolates of *B. bassiana*. Although four of them were able to produce siderophores, isolates RGM-731 and RGM-644 highlighted by their high siderophores exudation capacity, 73% and 81%, respectively. Our results show the capacity of IPF to solubilize nutrients at the isolate-specific level, which contributes to our knowledge of these fungi and their function as plant growth promoters.

In higher plants two different strategies have been described; Strategy I which includes all plants except grasses and Strategy II that it is confined to grasses; dicots or Strategy I, is characterized by the necessity to reduce Fe^3+^, to Fe^2+^, prior to its absorption, this reduction is mediated by a ferric reductase located in the plasma membrane of the epidermal root cells codified by *FRO2* gene in *Arabidopsis thaliana*. Once Fe^3+^ has been reduced, it is transported into the cells by a Fe^2+^ transporter codified by *IRT1* in *A. thaliana* [13,19,34,76,77,78,79,80,81]. Some plants species also induce H+ -ATPases responsible for rhizosphere acidification [54]. This work shows for the first time a role of an IPF as elicitor of the Fe deficiency responses in Strategy I plants. However, in the bibliography it can be found some examples of microorganisms e.g., bacteria and fungi, that induce Fe deficiency responses, ferric reductase activity and relative expression of the Fe acquisition genes. Some genera of saprophytic, phytopathogenic fungi, including mycorrhizae, such as *Paelomyces*, *Aspergillus*, *Penicillium*, *Gliocladium*, *Trichoderma*, *Gongronella*, *Fusarium*, among others, have been recorded as capable of solubilizing nutrients such as P and K [74,82,83]. Among them one of the most studied species is *Azospirillum brasilense*, cucumber plants inoculated with *A. brasilense* showed higher ferric reductase activity and relative expression of the Fe acquisition genes, *FRO1, IRT1, FIT, HA1* and *FRO3* [84,85]. Similar results were obtained in *A. thaliana* plants inoculated with *Bacillus subtilis* and *Pseudomonas simiae* [40,86]. Relative to the fungus species we found arbuscular mycorrhizal [87,88], *Trichoderma asperellum* and *Trichoderma harzianum* [89,90]. Recently, Lucena et al. [91] found that two yeast strain, *Debaromyces hansenii* and *Hansenula polymorpha* were able to induce Fe deficiency responses in cucumber plants. However, any works relative to IPF as Fe deficiency responses inductor can be found in the literature.

In this work the ability of *M. brunneum* 01/58-Su strain to induce Fe deficiency responses have been studied in two Cucurbitaceae species, C. sativus and C. melo. The results obtained show that *M. brunneum* 01/58-Su strain clearly induced the Ferric reductase activity and the relative expression of the Fe acquisition genes, *FRO, IRT1, HA1* and *FIT* in both species. These new skills of *M. brunneum* 01/58-Su strain confer him an added value to its use as an excellent biological control agent and highlight the direct and indirect mechanisms involved in the Fe acquisition as mediated by an IPF.

## Figures and Tables

**Figure 1 jof-09-00258-f001:**
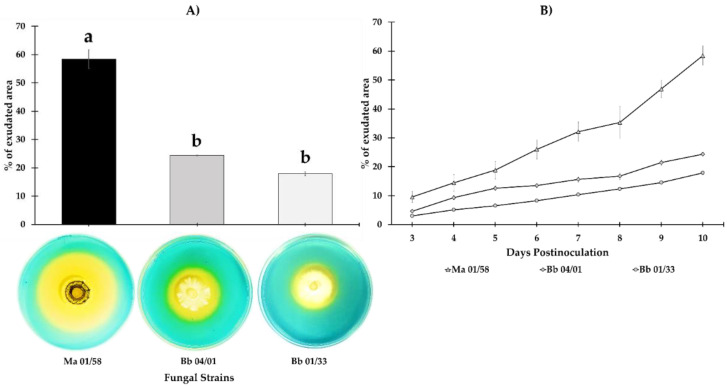
Siderophore exudation by three isolates of IPF on CAS agar medium with FeCl_3_ as FeIII source. At the bottom, the front of plates is shown. (**A**) Comparison at 10 days post inoculation (dpi). Bars with different letters are significantly different to each other according to Tukey test (*p* < 0.05). (**B**) Progress (%) of colour change due to siderophore production by three isolates of IPF on CAS agar medium.

**Figure 2 jof-09-00258-f002:**
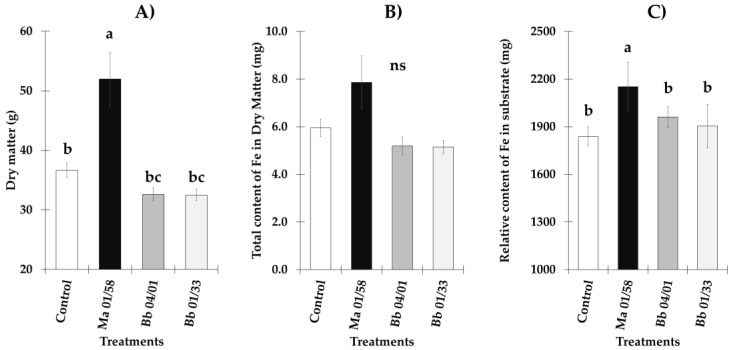
Mean (±SE) of stem and leaves dry matter weight (**A**), total content of Fe in dry matter (**B**), relative content of soluble Fe in substrate (**C**) measured at 50 dpi in melon plants inoculated by soil drenching. Letter over the bars denote significant difference between inoculated and control plants analyzed by completely randomized ANOVA followed by a Tukey test (*p* < 0.05).

**Figure 4 jof-09-00258-f004:**
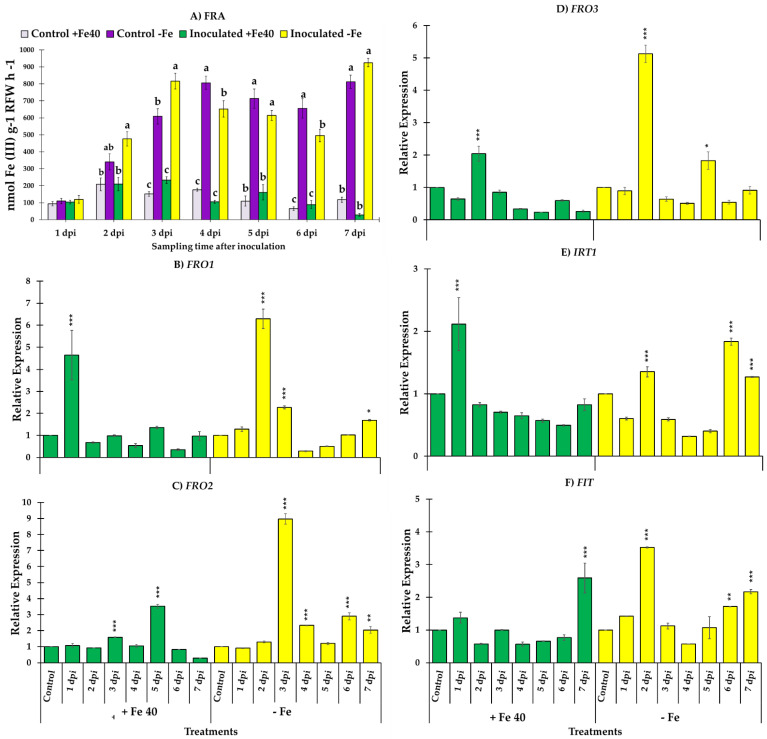
Evolution of FRA along seven days of observation and relative expression of *FRO1*, *FRO2*, *FRO3*, *IRT1* and *FIT*, in *C. melo* roots. Four treatments were used, namely, (i) Control + Fe40µM (un-inoculated), (ii) Inoculated + Fe40µM, (iii) Control - Fe (un-inoculated) and (iv) Inoculated - Fe. The expression of control treatment for each nutritional condition is presented once, at the beginning of the graph, with the relative expression comparison method used, the control is always equal to 1. Data of gene expression represent the mean of three independent technical replicates, according to the Dunnett’s test, * (*p* < 0.05), ** (*p* < 0.01) or *** (*p* < 0.001) over the bars indicate significant differences in relation to the control treatment. In the case of FRA, letter over the bars denote significant difference between plants inoculated and control plants analyzed by completely randomized ANOVA followed by a Tukey test (*p* < 0.05).

**Figure 5 jof-09-00258-f005:**
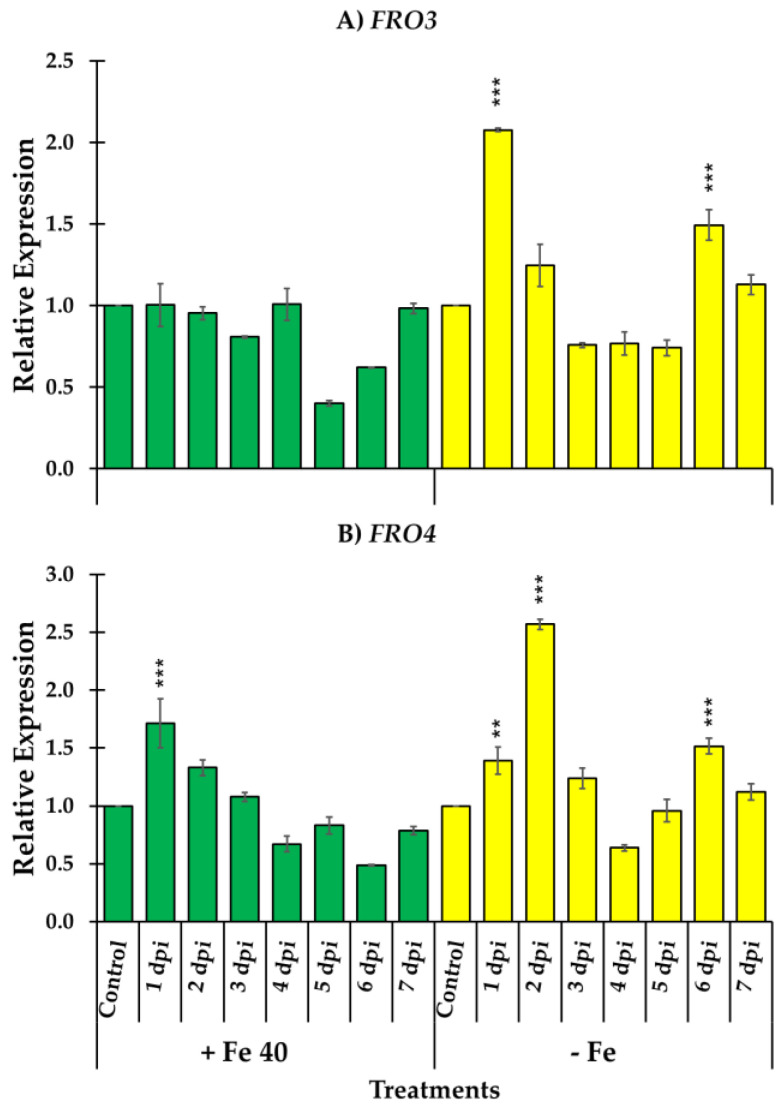
Relative expression of *FRO3* (**A**) and *FRO4* (**B**) on shoots of *C. melo*. Four treatments were used, namely, (i) Control + Fe40µM (un-inoculated), (ii) Inoculated + Fe40µM, (iii) Control - Fe (un-inoculated) and (iv) Inoculated - Fe. The expression of control treatment for each nutritional condition is presented once, at the beginning of the graph, with the relative expression comparison method used, the control is always equal to 1. Data of gene expression represent the mean of three independent technical replicates, according to the Dunnett’s test, * (*p* < 0.05), ** (*p* < 0.01) or *** (*p* < 0.001) over the bars indicate significant differences in relation to the control treatment.

**Figure 6 jof-09-00258-f006:**
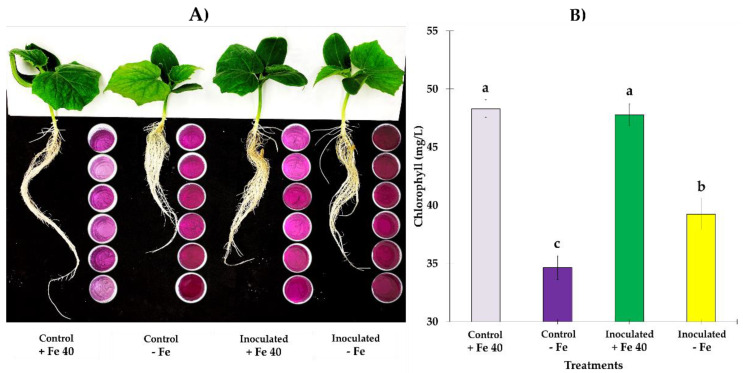
(**A**) General panorama of FRA on roots of *C. sativus*. On the right side of the roots, the indicator solution containers can be seen at 5 dpi, the FRA is generally highly induced, however, as can be seen in the intense purple color, inoculated plants exceed their respective control; shoots of inoculated plants did not show sever symptoms of chlorosis like occurred in plants without inoculation. (**B**) Mean of SPAD values from at 7 dpi showed significant difference between control and inoculated plants grown in Fe deficient conditions, exceeding inoculated plants their respective control. Letter over the bars denote significant difference between plants inoculated and control plants analyzed by completely randomized ANOVA followed by a Tukey test (*p* < 0.05).

**Figure 7 jof-09-00258-f007:**
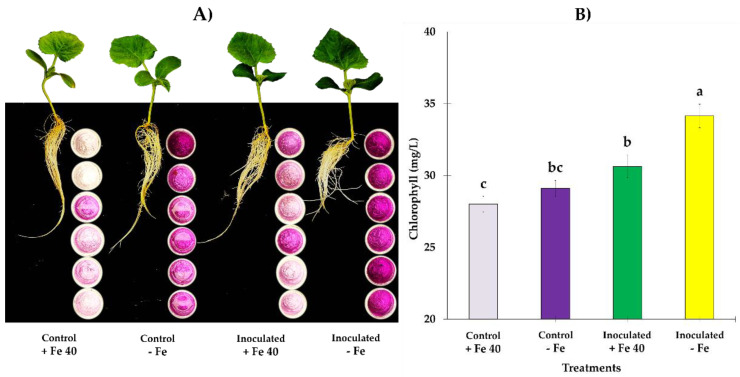
(**A**) General panorama of FRA on roots of *C. melo* at 5 dpi. On the right side of the roots, the indicator solution containers can be seen at 5 dpi with clearly significant difference between controls and inoculated plants. Also, abundant secondary roots growth can be observed; in shoots of inoculated plants did not show chlorosis symptoms. (**B**) Mean of SPAD values at 7 dpi revealed significant difference with inoculated plants exceed their respective controls, being the plants grown in Fe deficient conditions those that reached higher values in chlorophyl content. Letter over the bars denote significant difference between plants inoculated and control plants analyzed by completely randomized ANOVA followed by a Tukey test (*p* < 0.05).

**Table 1 jof-09-00258-t001:** Fungal isolates used in experiments.

Isolate	Fungal Species	Origin	Agroecosystem	Habitat	GenBank Accession Number	Spanish Type Culture Collection Accession Number
EABb 04/01-Tip	*B. bassiana*	Ecija (Sevilla, Spain)	Opium poppy crop	Insect (*Iraella luteipes*)	FJ972963	20744
EABb 01/33-Su	*B. bassiana*	El Bosque (Cadiz, Spain)	Traditional olive orchard	Soil	FJ972969	21149
EAMa 01/58–Su	*M. brunneum*	Hinojosa del Duque (Córdoba, Spain)	Wheat crop	Soil	JN900390	20764

**Table 2 jof-09-00258-t002:** Primers used in qRT-PCR analysis.

Gene	Gene Function/Name	Accession No.	Reference	Sequence	Species	Tissue
*FRO1*	Ferric reductase oxidase	AY590765	[61]	F: ATACGGCCCTGTTTCCACTT R: GGGTTTTGTTGTGGTGGGAA	*C. sativus*	Roots
*FRO1*	Ferric reductase oxidase		[62]	F: TCACAGCGATTTAGAACCAGA R: GCCTTCGAGGGAAACTTGAA	*C. melo*	Roots
*FRO2*	Ferric reductase oxidase		[62]	F: TCTATCTAATCCATGTGGGAGTAGC R: AACAGCGCCAGAAGGAAGAT	*C. melo*	Roots
*FRO3*	Ferric reductase oxidase		[62]	F: CGAAGGCTGAAGTATAAACCAAC R: ACCTTGTCCATGACTCATCACA	*C. melo*	Roots/Shoots
*FRO4*	Ferric reductase oxidase		[62]	F: CACCGTCGAATTGGTCCT R: TGGACTCGACGACACACTGAA	*C. melo*	Roots/Shoots
*IRT1*	Iron-Regulated Transporter1	AY590764	[61]	F: GCAGGTATCATTCTCGCCAC R: ATCATAGCAACGAAGCCCGA	*C. sativus*	Roots
*IRT1*	Iron-Regulated Transporter1		[62]	F: ATCCCAATGTTGCACCCGGATAGA R: AAACCGGTGGCGAGAATGATACCT	*C. melo*	Roots
*HA1*	ATPase	AJ703810	[61]	F: GGGATGGGCTGGTGTAGTTTG R: TTCTTGGTCGTAAAGGCGGT	*C. sativus*	Roots
*FIT*	Induced Transcription Factor		[62]	F: GACATCAACGATCAATTTGAG R: CGATCCTCGATCAAGCAA	*C. melo/C. sativus*	Roots
*Actin **	Actina	XM_004136807	[61]	F: AACCCAAAGGCAAACAGGGA R: TCCGACCACTGGCATAGAGA	*C. melo/C. sativus*	Roots/Shoots
*Cyclo **	Cyclophilin	NM_001280769	[61]	F: ATTTCCTATTTGCGTGTGTTGTT R: GTAGCATAAACCATGACCCATAATA	*C. melo/C. sativus*	Roots/Shoots

* Reference genes.

## Data Availability

The data from the present study are in the possession of the authors and are available for consultation upon request.

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
