# Peer review of "Entomopathogenic Fungi-Mediated Solubilization and Induction of Fe Related Genes in Melon and Cucumber Plants"

_jof, 2023, doi:10.3390/jof9020258_

Round 1
Reviewer 1 Report (Previous Reviewer 2)
My concerns have been adequately considered, and the manuscript has been significantly improved. I think it is acceptable now.
Reviewer 2 Report (Previous Reviewer 3)
Have no comments.
This manuscript is a resubmission of an earlier submission. The following is a list of the peer review reports and author responses from that submission.
Round 1
Reviewer 1 Report
The work is interesting, abording the regulation of Fe, P and K by entomopatogenous fungi, with projection of use in agriculture, for example. The work has several mistakes in the redaction, specially in the punctuation. I attach the pdf puntualizating several mistakes.
The figure 1 is not necessary.
The authors never explain the use of dpi.
The references are not homogeneus. 60 references are too many.

Author Response
Please see the attachment.
RESPONSE TO REVIEWER 1:
Reviewer 1:
The work is interesting, abording the regulation of Fe, P and K by entomopatogenous fungi, with projection of use in agriculture, for example. The work has several mistakes in the redaction, specially in the punctuation. I attach the pdf puntualizating several mistakes.
Dear reviewer, thank you very much for your opinion and valuable review, we appreciate them. All the general mistakes in the attached pdf have been addressed. Also, after carrying out the corrections according to your suggestions and comments, the complete work was sent to a native English-speaking for respective corrections (J.K. Pell Consulting).
The figure 1 is not necessary.
Done, we removed the figure 1.
The authors never explain the use of dpi.
Done. See line 20 and line 115 of the new version of the manuscript.
The references are not homogeneus. 60 references are too many.
Done, we checked and homogenized all the references. Also, we reduced the number.

Reviewer 2 Report
García-Espinoza et al. sought to evaluate the potential activities of entomopathogenic fungi in assimilation of minerals. This work is very interesting and deserves further investigation which will facilitates crop production in the sustainable agriculture. The manuscript was written well; however, the research results are too preliminary to be recommended to the journal JoF.
Major suggestions:
1. The whole research is too preliminary, including experiment design and research method (e.g., detection of siderophore).
2. In all assays, conidia, not mycelia, should be suggested as initial inocula. Due to the difference in vegetative growth among strains, using mycelia as inocula will resulted in some errors.
3. All figures should be re-organized to display the results concisely.
4. Please optimize the title for manuscript; in particular, the ‘regulation’ resulted in some confusion.
Minor suggestions:
1. Line127-128. Please correct the molecular formula listed as the ingredients in medium.
2. Line 133. Change ‘halozone’ into ‘halo zone’.
3. Line130. ‘discs of 6 mm of mycelium’ should be defined at the first use. In following text (line 140), the abbreviation should be used and make the text concise.
4. It is suggested that Figure1 should be moved to supplementary information.
5. Line 175. There must be a space between ‘3’ and ‘dpi’.
Author Response
Please see the attachment.
RESPONSE TO REVIEWER 2:
Reviewer 2
Major suggestions:
- The whole research is too preliminary, including experiment design and research method (e.g., detection of siderophore).
- In all assays, conidia, not mycelia, should be suggested as initial inocula. Due to the difference in vegetative growth among strains, using mycelia as inocula will resulted in some errors.
- All figures should be re-organized to display the results concisely.
- Please optimize the title for manuscript; in particular, the ‘regulation’ resulted in some confusion.
Dear reviewer, thank you very much the valuable review, we appreciate it and we sure that will contribute to improve the whole manuscript. In relation to your comments, we would like to clarify some aspects about these 4 comments:
We understand your first observation, but however, you should consider that in a such new field, the preliminary studies are very necessary to understand mechanisms and to build a solid advanced knowledge later. Many studies have addressed the role of entomopathogenic fungi as plant growth promoters but few, very few, have studied the mechanisms of this growth promotion at physiological and molecular level. In this sense, our objective is to provide a complete understanding to this phenomenon by series of papers, this is the first one. Also, the production of siderophores and solubilization of mineral nutrients have been widely studied in microorganisms such as bacteria, however, in entomopathogenic endophytic fungi this issue has not been fully addressed, in this sense, it is possible to contribute to the knowledge of the abilities of these fungi to promote plant growth. Moreover, this study was based on published scientific methodology proposed by Barra-Bucarei et al. (2020), who propose the use of mycelium of 4 days of growth; in addition, the works of Nautiyal (1999), Premono et al. (1996), Schwyn and Neilands (1987) as well as Andrews et al. (2016), all of them in the references of the manuscript, were taken as a reference for the quantification and measures related to the demineralization of the nutrients studied.
Finally, figures were re-organized, and title was updated. All minor suggestions were attended, and as other reviewer suggested, figure 1 was removed from the manuscript.
The complete work was sent to a native English-speaking for respective corrections (J.K. Pell Consulting).
Minor suggestions:
- Line 127-128. Please correct the molecular formula listed as the ingredients in medium.
Done. See lines 124-125 of the new version of the manuscript.
- Line 133. Change ‘halozone’ into ‘halo zone’.
Done.
- Line 130. ‘discs of 6 mm of mycelium’ should be defined at the first use. In following text (line 140), the abbreviation should be used and make the text concise. Done. Line 112 of the new version of the manuscript.
- It is suggested that Figure1 should be moved to supplementary information.
As we previously mentioned, figure 1 has been removed from the manuscript as reviewer 1 suggested.
- Line 175. There must be a space between ‘3’ and ‘dpi’.
Done, line 168 of the new version of the manuscript.

Reviewer 3 Report
This study investigated the capacity three endophytic insect pathogenic fungi strains for their ability to demineralize nutrients such as Fe, K, and P with strain-specific capacity. Results showed that strain EAMa 01/58-Su of Metarhizium brunneum showed the highest iron siderophore production. And EABb 01/33-Su of Beauveria bassiana has the highest indices of phosphate solubilization. EABb 04/01-Tip strain exhibit a higher K-solubilization. I provided some comments for the authors to consider as outlined below.
1. Abstract is need to be extended by using more significant results
2. Some phrases in Introduction were not clear and need to rewrite, for example, Line 50:… phosphate (P) and Potassim (K); Line 51-52: there is lack of knowledge respect fungi…; The third paragraph and the fourth paragraph can be merged.
3. All methods need to be addressed.
4. In my opinion results of this study are sufficient but, I recommend to clarify and describe more. The main problem was the quality of figures. Figures of 2 and 3, Figures of 4 and 5, Figures of 6 and 7, and Figures of 8 and 9 should be merged, respectively.
5. The number of references should be cleared, especially in the second paragraph of Discussion.
6. To determine the capacity for siderophores production, solubilization of P and K of two genera Beauveria and Metarhizium, in my opinion, more strains of Beauveria Metarhizium should be investigated.
Author Response
Please see the attachment.
RESPONSE TO REVIEWER 3:
Reviewer 3
This study investigated the capacity three endophytic insect pathogenic fungi strains for their ability to demineralize nutrients such as Fe, K, and P with strain-specific capacity. Results showed that strain EAMa 01/58-Su of Metarhizium brunneum showed the highest iron siderophore production. And EABb 01/33-Su of Beauveria bassiana has the highest indices of phosphate solubilization. EABb 04/01-Tip strain exhibit a higher K-solubilization. I provided some comments for the authors to consider as outlined below.
Dear reviewer, thank you very much for your opinion and valuable review, we appreciate them. We have addressed all your comments. Also, the complete work was sent to a native English-speaking for respective corrections (J.K. Pell Consulting).
- Abstract is need to be extended by using more significant results.
Done, we added some results to the abstract. However, due to the limit of wards (200), it will be difficult to address more.
- Some phrases in Introduction were not clear and need to rewrite, for example, Line 50:… phosphate (P) and Potassim (K); Line 51-52: there is lack of knowledge respect fungi…; The third paragraph and the fourth paragraph can be merged.
Done. As we previously mentioned, the whole manuscript has been checked and corrected by Prof. Judith Pell consulting to address any lack of language mistakes.
- All methods need to be addressed.
We think that we addressed all methods. Also, after considering the comments and corrections of the other two reviewers, the manuscript became more complete and easier to understand.
- In my opinion results of this study are sufficient but, I recommend to clarify and describe more. The main problem was the quality of figures. Figures of 2 and 3, Figures of 4 and 5, Figures of 6 and 7, and Figures of 8 and 9 should be merged, respectively.
Done, we addressed this point by merging the figures as indicated.
- The number of references should be cleared, especially in the second paragraph of Discussion.
Done, we checked all the references and reduced the number.
- To determine the capacity for siderophores production, solubilization of P and K of two genera Beauveriaand Metarhizium, in my opinion, more strains of Beauveria Metarhizium should be investigated.
Dear reviewer, you are absolutely right about including more strains. In our future studies, we will include some screenings using more strains of our collection. However, in this study, our main objective was to deepen the knowledge of the mechanisms used by these strains to promote plant growth since in our previous studies these strains were the best at colonizing plants, controlling piercing-sucking and chewing insects and promoting plant growth.

Round 2
Reviewer 2 Report
The manuscript has not significantly improved.
Author Response
Dear reviewer
We would to thank you again for the 2nd round of the revision witch undoubtedly improved the manuscript redaction and understanding. In this letter, we would to explain some pints regarding your revision.
First of all, we improved the introduction section by adding more information about the demineralization of nutrients by other microorganisms such as bacteria of fungi (not insect pathogenic). Please take into account the novelty of the work that address the role of three well known (very studied as insect pathogenic) strains of entomopathogenic fungi to solubilize nutrientes and the difference between strains in doing that.
About the research methods, as we mentioned in our previous revision, we used the latest methods published by Barra-Bucarei et al (2020) and other published works. However, we will of course take into account your comments regarding that in our future studies since it would be interesting to study the same with with mycelium instead of conidia or to compare with conidia.
Dear reviewer, our conclusion was very simple, literally, we said: Our results show the capacity of IPF to solubilize nutrients at the isolate-specific level, which contributes to our knowledge of these fungi and their function as plant growth promoters. We think that this study will contribute to the knowledge about the role of IPF as plant growth promoters.
Many thanks
The authors